# Dynamic Characteristics and Failure Mechanism of Vegetated Revetment under Cyclic Loading

**DOI:** 10.3390/ma12050716

**Published:** 2019-03-01

**Authors:** Wei Chen, Ruoyu Jin, Han Zhu, Yidong Xu, Dariusz Wanatowski, Lili He, Qinglin Guo

**Affiliations:** 1School of Civil Engineering, Tianjin University, 92 Weijin Road, Tianjin 300072, China; hanzhu2000@tju.edu.cn; 2School of Civil Engineering & Architecture, Ningbo Institute of Technology, Zhejiang University, 1 Xuefu Road, Ningbo 315100, China; xyd@nit.zju.edu.cn; 3School of Environment and Technology, University of Brighton, Cockcroft Building 616, Lewes Road, Brighton BN2 4GJ, UK; 4Faculty of Engineering, School of Civil Engineering, University of Leeds, Leeds LS2 9JT, UK; d.wan@leeds.ac.uk; 5School of Civil Engineering, Hebei University of Engineering, Handan 056038, China; helili@hebeu.edu.cn (L.H.); guoql@hebeu.edu.cn (Q.G.)

**Keywords:** cyclic loading, vegetated bituminous mixture, dynamic characteristics, failure mode, numerical simulation

## Abstract

This research is focused on the dynamic behavior and failure mechanisms of an ecologically vegetated bituminous mixture applied in a riverbank revetment model. The dynamic bearing capacity of the vegetated riparian slope was evaluated. The dynamic soil pressure distribution and deformation were analyzed, followed by 3D elastic–plastic finite element modeling. Experimental results showed that the cumulative vertical settlement increased rapidly with the loading time. Vegetation added into bituminous mixtures was found to be effective in inhibiting the development of the vertical displacement of sand. The research described in this paper provides a theoretical basis and guidelines for the protection of riverbank slopes.

## 1. Introduction

With the rapid urbanization movement in China, urban ecological friendliness is becoming an emerging issue. Ecological systems including vegetation and plants around riverbanks have been damaged in some areas of China, leading to potential slope failure, and a series of environmental problems such as soil erosion, debris flow, and food chain destruction. These environmental problems have occurred in several coastal areas both in China and overseas and have aroused public attention [1,2,3,4,5,6,7,8]. A porous planting asphalt concrete or vegetated bituminous mixture could be applied in riparian revetments as one potential solution to mitigate these environmental problems. Ecological grass or plants can be incorporated into the bituminous mixture. Due to the widely distributed pores, moisture and air can easily enter a vegetated bituminous mixture. Growing vegetations in bituminous mixture could become a new technology. It is suitable for application in housing, municipal engineering and for the slope structures of urban river revetments, subgrades and pavement surfaces. It is also believed to be an effective method of solving existing environmental problems in riverbank revetments, especially given that the stability of riverbank revetments has become a hot research topic worldwide [9,10,11,12].

Some theoretical and experimental studies have been performed to target the engineering stability of riverbanks and for slope protection. For example, Van Beek and Bogaard [13] found that unstable soil was significantly stabilized when completely planted with vegetation. When vegetation was planted deep into the unstable soil, the enhancement of soil stability was generally limited to the soil surface [13]. Osman and Barakbah [14] identified the root density and soil moisture content as the slope stability index. It was further found that the planting root density was positively correlated with the soil shear strength, and the soil moisture content was negatively correlated to the stability of riverbank revetments [14]. Zhou and Zhao [15] found that the shear strength of soil on the revetment surface increased significantly by adding grass roots, and the cohesion was about 35%~100% higher than that of the revetment soil without grass roots. The stability coefficient of the shallow layers of the revetment increased by about 2%~11% due to the reinforcement effect from grass roots [15]. Zhang et al. [16] focused on the mechanism of bonding between lateral roots and soil and developed a failure model of frictional root-soil bond. The shear tests performed by Zhang et al. [17] revealed that planting root density had a significant impact on the shear strength of soil. Basically, root density had a positive correlation with the soil shear strength before the density reached a certain limit [17]. Afterwards, the root density would cause the reduction of the shear strength of soil [17]. Ahmed and EI-Kourd [18] analyzed porous vegetated concrete slopes and suggested that surface cover materials should be able to resist rain erosion. Malhotra and Carette [19] developed a slope protection technology to prevent planting from flooding and to mitigate the difficulty of restoring moisture within planting. Zhou and Chen [20] found that the linear viscoelastic behavior of the asphalt concrete was characterized by the generalized Maxwell model according to the results of dynamic modulus test. Si and Cao [21] studied the stress and deformation distribution characteristics of HMAC (high-modulus asphalt concrete) pavements and found that under the action of moving loads, the strain and displacement generated in HMAC pavements were lower than that in conventional asphalt pavements. Bazzaz et al. [22] indicated that a straightforward procedure method was capable of characterizing the NVE (nonlinear viscoelastic) response of asphalt concrete materials.

A review of the existing literature indicated that research into the stability of riparian revetment adopting vegetated bituminous mixtures is mostly limited to static loading conditions [23,24,25,26,27]. Existing review-based studies [28] have not sufficiently addressed the dynamic features of bituminous mixtures, especially under cyclic loading [29]. However, studying the dynamic behavior of bituminous mixtures is important, especially for those applied in transportation infrastructure under high traffic loads [30]. The existing studies of sustainable bituminous mixtures [31,32,33] have not sufficiently addressed their ecological friendliness when applied in riverbank structures. On the other hand, the riparian revetment will be inevitably subjected to cyclic loads such as rainwater scouring, rolling, and adjacent construction [34]. The failure mode of revetments under dynamic cyclic loading has not been widely studied.

In the current study, a physical riverbank revetment facility was built to represent the real-world scenario of the failure progress of a riverbank revetment. Adopting reasonable ratios of heterogeneous raw materials, the vegetated bituminous mixture samples were obtained for testing in the natural gravity environment. The revetment test facility was subjected to dynamic cyclic loading. The experimental tests were designed to analyze the revetment’s dynamic bearing capacity, deformation, and dynamic soil pressure. Afterwards, the experimental test results were compared to a numerical simulation adopting a three-dimensional elastic-plastic finite element model (FEM), in order to explore the failure mechanism of the vegetated riverbank revetment under long-term cyclic loading. 

## 2. Test Schemes

In order to investigate the dynamic characteristics of vegetated riparian revetment under cyclic loading, the control sample, made of unvegetated bituminous mixture, was prepared as the comparison to the bituminous mixture with grown plants applied in the riverbank revetment. The bituminous mixture layer had a thickness of 60 mm, which was considered to be suitable for growing vegetation.

### 2.1. Vegetated Bituminous Mixture

The purpose of studying the vegetated bituminous mixture was to enable the plants or grass to grow within bituminous materials, as a method for soil-less cultivation. To enable vegetation in the porous bituminous mixture, it is critical that raw materials and vegetation growth are present before applying the vegetated bituminous mixture to the revetment facility. 

#### 2.1.1. Raw Materials

The vegetated bituminous mixture was prepared using multiple raw materials including SBS (i.e., Styrene-Butadiene-Styrene) modified bitumen, vermiculite powder (absorption rate between 18% and 20% and the expansion ratio from 10 to 20), perlite powder (density of 70 kg/m^3^ and compressive strength of 0.5 MPa), sodium carboxymethyl cellulose (CMC-Na), soilless cultivation nutrient solution, ferrous sulfate, soil nutrition, organic compound fertilizer, and Setaria. 

The SBS modified bitumen was made by adding SBS modifier into the original bitumen. SBS modifier was evenly dispersed in the bitumen by shearing and stirring. A stabilizer was also added into the SBS bitumen to form an SBS blend. The original bituminous mixture was firstly compressed into desired forms using machines in the factory. The compressed and shaped mixture samples were then cooled down before the porous asphalt was infiltrated with the cultivation solution. More detailed steps of growing vegetation and preparing the vegetated bituminous mixture are described below:Adopting an appropriate proportion of nutrient solution, ferrous sulfate particles and water to form a mixed solution;Properly mixing vermiculite powder, perlite, nutritive soil, compound fertilizer and seeds to prepare the cultivation environment for vegetation;Adding CMC-Na thickener into the cultivation solution until it reached a uniform state;Finally, infiltrating the viscous cultivation solution from the last step into the porous bituminous mixture; an alternative approach would be to immerse the porous bituminous mixture into the cultivation solution for half an hour (see Figure 1). The cultivation solution should be abundant to immerse the asphalt samples. Theoretically, the porous asphalt has a porosity rate of around 20% and could be fully filled with cultivation solution. In practice, however, volumes between 10% and 15% of asphalt samples were filled by cultivation solution.

#### 2.1.2. Vegetation Preparation and Growth

Following the raw material preparation and planting procedure, the maintenance work became critical, including watering the vegetation, fertilizing, and removing impurities. Generally, the vegetation was watered twice or three times per day, and a higher frequency of watering would be required in hot weather to ensure the growth of vegetation roots. The growth status of vegetation should also be watched to ensure the proper amount of fertilization. Generally, fertilization once every ten to 14 days would be sufficient. It was also recommended to mix the nutrition solution and ferrous sulfate into a diluted solution to water the vegetation. During the growth period, plastic film covering was also needed to protect vegetation from low temperature on cold days. The growth of vegetation at different cultivation days is illustrated in Figure 2.

It can be seen from Figure 2 that Setaria grew throughout the cultivation days, with its roots growing in both size and quantity. On day 33, fibrous roots were also observed to be growing beside the main root. A longer cultivation period would lead to a more developed root system of the vegetation.

The layering method was applied to set the thickness of 30 cm for each soil layer to ensure that the surface layer would achieve the required density and moisture content. The experimental revetment model was constructed by specified forming devices. After the revetment was formed, water was added until the saturated state was reached. The saturation rate was measured using the electrical resistivity method. After stalling for a certain period of time, water was released.

Once the revetment was formed and properly saturated, the vegetated mixture was placed evenly on the revetment surface. Continuous irrigation was applied to nurture the vegetation roots to grow downwards until they were fully consolidated with the foundation soil. Two main steps were implemented to ensure the full consolidation of vegetation with the foundation soil. Firstly, an asphalt concrete mixture was added with a mineral admixture (i.e., CMC-Na thickener) to increase its internal porosity and allow more internal space for vegetation growth. The pH value was also reduced to meet the planting growth requirements. The local climate was considered to select proper vegetation species, and proper seeding and cultivation methods were adopted by observing and evaluating the vegetation growth. More details of the vegetation nurturing and growth can be found in the researchers’ earlier publication [35]. The growth and nurturing of vegetation in asphalt concrete was also tested and evaluated in the earlier research [35]. The details of the vegetated bituminous mixture are presented in Table 1. 

Secondly, the evaluation of the consolidation of vegetation within foundation soil was based on the utilization of a scale dynamometer to determine the maximum resistance of plant roots at test failure. A vernier calliper was used to measure the root fracture surface diameter. Then, the tensile strength of the test root sample would be calculated using Equation (1):
(1)$$P=4F/\pi D^{2}$$
where *F* denotes the maximum resistance at root failure, and *D* is the diameter of the fracture surface. Up to 28 trails were performed for the test of consolidation between the vegetation and foundation soil. 

### 2.2. Physical and Mechanical Properties of Foundation Soil 

In order to accurately simulate the actual condition of riparian revetment, the sand and soil from the local coastal areas were collected and used for the experimental study. Sieve analysis was conducted to define the particle size distribution. According to the high-pressure direct shear test (results shown in Figure 3b), the cohesion was identified at 2.2 kPa and the internal friction angle *φ* was determined as 34.5°. Consistent geotechnical tests (i.e., high-pressure direct shear tests, as shown in Figure 3a) were conducted on the bituminous mixture. The physical and mechanical properties of the foundation soil and the bituminous mixture are summarized in Table 2. These geotechnical parameters are effective for numerical modelling afterwards.

The compaction index and the void ratio of the tested materials were 0.95 and 0.85, respectively, for the later physical experimental models (i.e., unvegetated and vegetated slopes).

### 2.3. Loading Device and the Monitoring Scheme

The dynamic triaxial cell supplied by GDS Instruments from the UK was adopted in this research to apply the static load or semi-sinusoidal cyclic load. The type of load was controlled by air pressure. The vertical load could be measured by the pressure sensor connected to the jack. The vertical displacement could be measured by the deformation gauge connected between the jack and the loading head. Another displacement transducer (i.e., LVDT or Linear Variable Differential Transformer) was used to record the lateral deformation. A soil pressure box was used to record the additional soil pressure of the revetment slope. When the inclination of the slope is low, the soil on the revetment can be stabilized even in the unreinforced state. If the revetment is too steep, the stability of the soil will be poor, and a slide could easily occur. In this research, the test facility shown in Figure 4 with a slope of 35° was designed and built. The schematic diagram of the test facility is shown in Figure 4.

According to Figure 4, the slope of the embankment considered in this study was 35°. Sand was used as the foundation soil. It should be noticed that erosion issues due to continuous irrigation supply were not considered in this study. Therefore, the erosion effects were excluded in the numerical modeling afterwards. The displacement transducer and cyclic loading facility are further shown in Figure 5. 

The dynamic load was simulated by the semi-sinusoidal wave. The amplitude of dynamic loading was set at 60% of the maximum bearing capacity (i.e., 78.8 kPa). The frequency was 1 Hz, and the number of cycles was equal to 10,000. The loading waveform is shown in Figure 6.

After the test facility was set up, the lateral LVDT was installed, and the loading plate was placed. The top of the loading plate was placed at the right center of the loading head. After the installation of all measuring instruments, the data acquisition software was also set. The loading process started from the zero-loading condition. During the loading process, the dynamic deformation of the revetment model was observed. Loading was stopped immediately when the deformation was increased, suddenly leading to revetment damage.

## 3. Analysis of Test Results

### 3.1. Displacement of Revetment

The settlements of the two physical models (i.e., unvegetated and vegetated models) of riverbank revetments were analyzed with the loading time. Figure 7 displays the vertical displacements of these two models. 

Figure 7 only demonstrates the vertical displacement during the first 20 s of the loading period. No rebound was observed after the initial 20 s test. Settlements after 20 s followed similar trends, as shown in Figure 7. It can be seen that at the beginning of the dynamic semi-sinusoidal loading, the two models displayed similar linear trends during which the soil was compacted. As the loading time increased, the settlements of the two models began to differ. The unvegetated revetment tended to have a significantly higher vertical displacement. The vegetation reduced the vertical displacement. The cumulative settlement of the riverbank revetment models is shown in Figure 8. 

Both of the two cumulative settlement curves in Figure 8 showed a linear trend until they reached their own maximum bearing capacity. There was then a critical or transitional point where the revetment models had a suddenly increased deformation leading to brittle failure. As a result, the sloped displacement curves shown in Figure 8 suddenly changed their directions. Figure 9 displays the displacement vector of slip surface at failure during the physical modeling using the PIV (particle image velocimetry) method.

The displacement vectors for both vegetated and unvegetated physical models are shown in Figure 9. The arrow indicates the displacement direction and its length shows the amount of displacement. Figure 9a displays the displacement vector of vegetated revetment, where the soil particles moved towards right and upwards during the test. Figure 9b is the failure of unvegetated revetment. It was found that the soil displacement of the vegetated revetment turned out to be larger than that of unvegetated revetment, indicating that the stress was diffused in the vegetated sample. The vegetated revetment had a more uniform soil stress distribution. 

The boundary conditions were based on two-way (i.e., horizontal and vertical) restrictions. The boundary constraint is a significant factor that influences the stability evaluation of revetment in finite element analysis. It significantly affects the safety factor and potential sliding surface within a revetment. The full constraint boundary creates the highest safety factor, around 30% higher than that in the free boundary condition [36]. Compared to the full constraint boundary, the semi-constraint boundary condition has a deeper sliding surface [36]. Considering the actual working conditions of this physical modeling and test, the fully constraint condition was set by applying horizontal and vertical restrictions. 

After 4000 cycles of loading, the settlement of the unvegetated revetment was around 5.43 mm before the failure occurred. In comparison, the vegetated revetment model failed after 8000 loading cycles with a maximum cumulative settlement of 12.85 mm. This indicates that the vegetation reinforced the revetment slope and mitigated its instability, extending its life two-fold.

### 3.2. Lateral Cumulative Deformation of Revetment

Three test points, namely the top of the revetment slope (Point C), the middle of the revetment slope (Point B), and the revetment slope crest (Point A), were selected for the test of lateral deformation, as shown in Figure 4. The three test points were instrumented with LVDT. The lateral deformation curve of each test point on the bank revetment is shown in Figure 10.

It should be noticed that a higher settlement in the vegetated revetment at 4000 cycles does not necessarily mean a reduction in stability. Instead, the vegetated revetment displayed a better stability. The higher settlement that occurred in the vegetated revetment was due to the lower overall stiffness of the revetment and foundation soil. It is seen from Figure 10 that all the three test points had higher cumulative settlement and underwent significantly more cycles of loading vibration due to the reinforcement effects from vegetation. Comparisons among the three test points show that the top position (i.e., point C) had a higher ultimate bearing capacity at 78.8 kPa and also higher lateral deformation, because the top position was closer to the loading center. 

According to Figure 10, after the maximum 4000 cycles of loading, the lateral deformation of the three test points (i.e., top, middle, and bottom) of the unvegetated revetment model reached 4.85 mm, 4.73 mm, and 4.21 mm, respectively. The vegetated revetment model underwent 8000 loading cycles, and the lateral deformations of the same three test points reached 14.35 mm, 15.22 mm, and 17.18 mm, respectively, representing increase rates of 196%, 221% and 303% compared to those in the unvegetated revetment slope. The vegetated bituminous mixture displayed a significant influence on the lateral displacement of the riverbank slope by increasing its overall stability.

### 3.3. Soil Pressure in Revetments under Dynamic Cyclic Loading 

For comparison purposes, three measurement points (i.e., E1, E2 and E3 as shown in Figure 4) were selected to measure the dynamic soil pressure. The three measurement points were in the same elevation, where E1 and E2 were right below the loading plate, and E3 was below the revetment slope. Figure 11, Figure 12 and Figure 13 demonstrate the dynamic soil pressures for the three measurement points. 

Figure 11, Figure 12 and Figure 13 also highlight the comparison between the unvegetated revetment and the vegetated model. It is seen that the soil pressure at E2, which was right below the loading center, was significantly higher than in E1 and E3. That was because E2 was closer to the loading position, with a denser soil and a higher pressure.

In the initial loading stage, each measurement point had a more fluctuating value of soil pressure; afterwards, the fluctuation gradually decreased and tended to stabilize to a constant value. This was because, in the initial test period, the soil underwent a fast compaction with a higher pressure. As the cyclic loading continued, the soil was softened. The settlement was increased and the soil pressure decreased. 

Table 3 summarizes the soil pressure at the three measurement points by comparing the vegetated and unvegetated conditions.

Compared to the unvegetated slope, the peak and mean values of soil pressure within the vegetated revetment were generally larger, indicating that vegetated bituminous mixtures could solidify the revetment slope and increase its stiffness.

## 4. FEM Analysis of the Stability of Vegetated Revetments

The physical test facilities in the indoor environment were further validated by comparing the experimental results to the FEM numerical simulation. The validation highlighted the effects of long-term cyclic loading on the lateral and vertical displacements, the soil stress distribution along the vertical direction, and the overall trend of revetment sliding.

### 4.1. Method of Applying Cyclic Loading

During the ABAQUS modeling, the custom load codes (i.e., DLOAD and VDLOAD) were used to define non-uniform distributed load. The load magnitude was the function of location, time and unit. The subroutine DLOAD was suitable for ABAQUS/standard computation, and VD-LOAD fitted ABAQUS/explicit computation. The cyclic load modeling in this study was based on implicit integration to solve the dynamic finite element equation, corresponding to dynamic/implicit in the ABAQUS program standard. Therefore, the load subroutine DLOAD was chosen [37].

### 4.2. Establishment of the Three-Dimensional Finite Element Model

To verify the experimental test results of the dynamic characteristics of the vegetated revetment under cyclic loading, the size of the riparian revetment model was consistent with the physical model displayed in Figure 4. The geotechnical parameters of the vegetated slope were directly incorporated into the finite element model (FEM). These geotechnical parameters were obtained from the prior physical tests, including the exact values of cohesion and internal friction angle shown in Table 2, with a Poisson’s ratio of 0.3. It should be noticed that only geotechnical parameters were incorporated in the FEM model. Hydraulic parameters were not considered. FEM could not exactly simulate the real-site conditions, although the stiffness of the vegetated river bank surface could be adjusted in the numerical modeling. In the numerical simulation, the X and Z axes represented the lateral directions, and the Y axis denoted the vertical direction. The three directional deformations along X, Y and Z axes were restrained. The surface of the revetment slope was set to allow sliding. All relevant attributes were assigned to the vegetated revetment, and then the cyclic loadings were applied to simulate and obtain the deformation and stress distribution of the revetment. The bottom of the slope was fully constrained, and horizontal constraints were applied to the left and right sides of the slope. The ideal elastoplastic constitutive model with the associated flow law was applied to the soil. The three-dimensional finite element model is shown in Figure 14.

### 4.3. Analysis of Numerical Simulation 

#### 4.3.1. Comparison between Test and Numerical Calculation

The comparison of cumulative settlement between experimental tests and numerical simulations for vegetated and unvegetated revetment models is shown in Figure 15. 

The settlements of the unvegetated revetment were 6.4 mm and 7.3 mm, respectively, according to the experimental and simulation results. The difference of settlement values between the experiments and FEM was 14.06%. The tested and simulated values of the settlement were 11.32 mm and 9.48 mm, respectively, for vegetated revetment, denoting a difference of 16.25%. The potential causes of the difference between FEM and experimental tests include (1) the edge effects within the physical model; and (2) the numerical modeling details such as the parametric selection and input, revetment material attributes, and the mesh partitioning details in FEM. It was further found that the results between the experimental tests and simulation tended to be closer as the number of loading cycles increased, indicating that the settlement values from FEM-based simulation were generally consistent with those from the experimental tests.

#### 4.3.2. Displacements along the Depth of Revetments

Figure 16 illustrates the vertical and lateral displacement values measured when instability of revetments occurred. The displacements were obtained along the depth right below the loading center, as shown earlier in Figure 4. 

It is found from Figure 16 that the vegetation played a significant role in restricting the vertical displacement along the whole depth under the loading center. However, the reinforcement effect of vegetation was more significant in the shallow depths of soil layers. The effect of vegetation decreased as the depth increased. Therefore, the vegetative technique has limited reinforcing effects on slopes prone to deep-level damage. Such a type of slopes should be first strengthened by other methods before the vegetation protection approach is adopted. 

The lateral displacement of the vegetated revetment was significantly different from that within the unvegetated model. When the slope depth was less than 150 mm, the lateral displacement of the vegetated revetment was significantly lower than that of the unvegetated model. The vegetated bituminous mixture effectively inhibited the lateral displacement. However, when the depth was more than 200 mm, the lateral displacements in vegetated and unvegetated models were similar. These observations suggest the lateral restraint effect of vegetation was more significant in the lower-depth layers.

#### 4.3.3. Sliding Trend of Vegetated Revetment 

The sliding trend of the vegetated revetment is demonstrated in Figure 17.

According to Figure 17, the vertical displacement right under the loading plate was the highest. The vertical displacement decreased along the depth below the loading center. Upon the occurrence of structural instability, the upper region of the revetment tended to slide outward. Under the cyclic loading, soil under the loading plate experienced a downward deformation. In the meantime, the two sides of the revetment were uplifted. The vegetation restricted the movement of the revetment surface and enhanced the stability. 

#### 4.3.4. Internal Stress Distribution in Revetments

The vertical stress along the depth of revetments is demonstrated in Figure 18.

The vertical stress of the unvegetated revetment gradually decreased along the depth. The transitional point of vertical stress in the soil occurred at around the depth of 100 mm, where the vertical stress decreased sharply. This was mainly due to the existence of the vegetated bituminous mixture in the structural layer, which diffused the vertical stress and changed the soil stress state. The strength of the shallow soil increased with the root content in the soil. 

It should be noticed that the effects of sulfate on soils were not investigated in this research. In fact, the effects of sulfate on soils are complicated according to existing studies [38,39,40]. They will be further studied in future research. The strength properties tested in this study were based on undrained conditions, as only short-term mechanical properties of revetments were analyzed. Due to the fact that drained conditions would prevail because of irrigation, future study should also consider the drained condition in order to further investigate the long-term properties of revetments.

## 5. Conclusions

This study adopted indoor physical models to test the dynamic performance and failure modes of riverbank revetments covered by vegetated bituminous mixtures. The dynamic characteristics of vegetated revetments were analyzed based on the model tests under cyclic loading conditions. The failure mode and dynamic behavior of vegetated revetments under cyclic loading were further verified using numerical simulation based on a three-dimensional finite element model. The main observations and findings are summarized as follows:As the loading cycle increased, accumulative settlement reached a critical point. Afterwards, the cumulative settlement increased sharply as a result of the revetment slope failure;The top of the revetment slope, which was closer to the loading center, there was a higher lateral deformation compared to the points at the middle and bottom of the slope;The vegetation in bituminous mixture had a significant influence on the lateral displacement of a revetment by increasing its overall stability;Within the same horizontal layer, the peak and mean values of soil pressure at the test point right below the loading center were significantly larger than the points on two sides;The maximum and mean values of soil pressure within the vegetated revetment were generally higher than those in the unvegetated models, indicating that the vegetated bituminous mixture increased the soil stiffness;The numerical simulation adopting a three-dimensional finite element model showed generally consistent vertical settlement values as the experimental tests did for both vegetated and unvegetated revetments. The potential causes of the differences between the numerical simulation and experimental tests were identified, such as the mesh partitioning in the finite element model;The vegetation had a reinforcement effect in restricting the vertical movement along the depth of the revetment under the loading plate. However, the reinforcement effect was more significant in the shallow soil layers.

The vegetated bituminous mixture added in the soil structural layers affected the soil stress distribution by diffusing the soil stress. The interaction between vegetation and foundation soil resulted in the stiffening effect on the riverbank revetment. This research provided the initial findings on revetment stability under cyclic loadings. It offered the guidelines to predict the stability of vegetated revetment in practice. 

The physical and numerical modeling in this study did not consider other influencing factors on riverbank conditions, such as waving effects on erosion, the changing soil–water pressures, or saturation conditions during the high levels of open water. Future tests of dynamic characteristics of vegetated revetment could extend the current research findings by introducing these influence factors. For example, it is known that, under the condition of soil saturation, with the increase of water content, the soil cohesion and internal friction angle of the phyto-sanitary layer would decrease. However, the embedded effect of vegetation roots still exists to protect the river bank slope by constricting the horizontal displacement. Whether and how the embedding effect will change due to soil saturation require further research. Another recommended direction for future work would be to investigate the plant root’s influence on the vegetated bituminous mixture by considering the long-term stiffness, root density, and the aging bituminous part of the mixture. 

## Figures and Tables

**Figure 1 materials-12-00716-f001:**
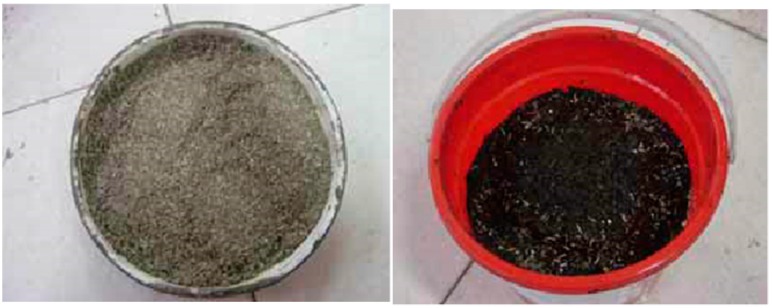
Mixing cultivation solution with the porous bituminous mixture.

**Figure 2 materials-12-00716-f002:**
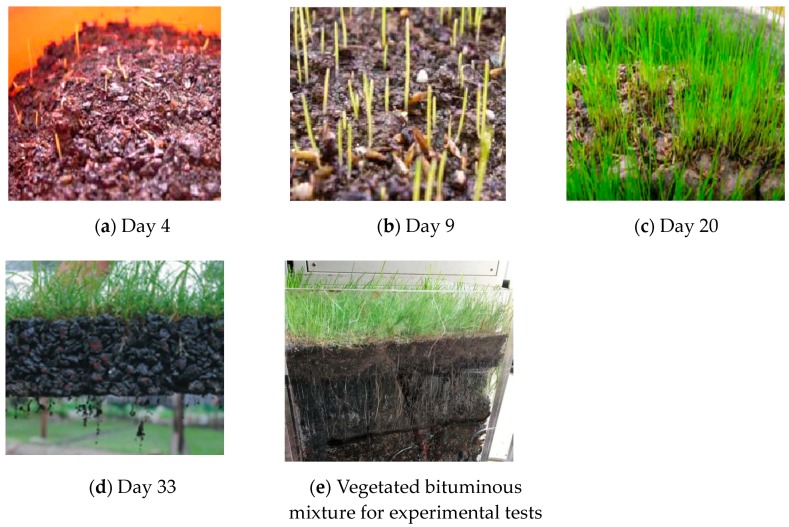
The growth of vegetation in bituminous mixture

**Figure 3 materials-12-00716-f003:**
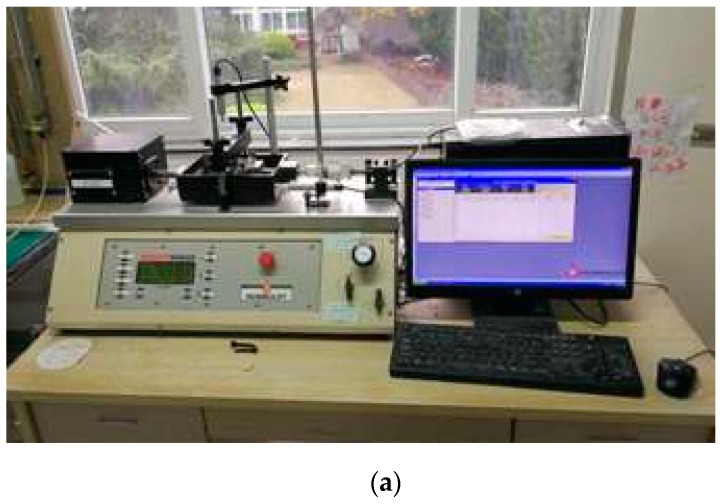
Shear test facility and test results: (**a**) high-pressure direct shear test facility; (**b**) direct shear test curves of foundation soil.

**Figure 4 materials-12-00716-f004:**
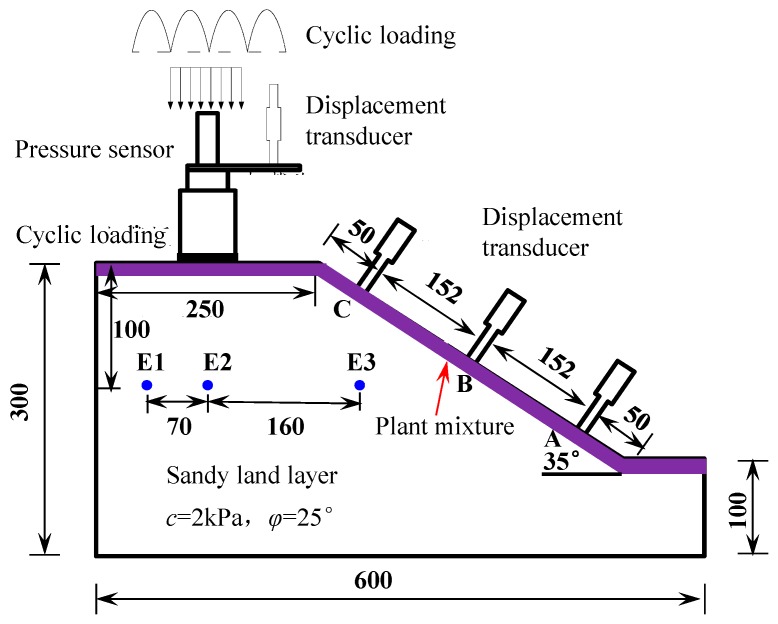
The schematic diagram of the test facility with various measurement instruments (unit: mm).

**Figure 5 materials-12-00716-f005:**
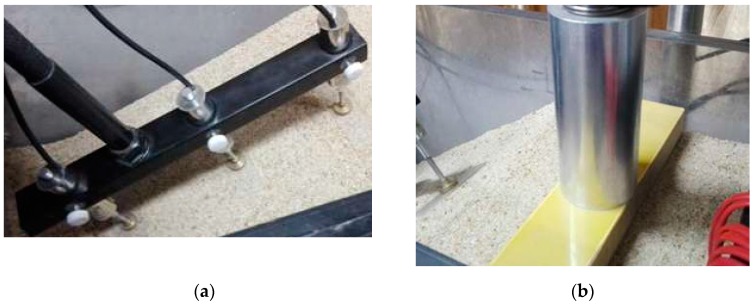
Details of loading and test facilities: (**a**) displacement transducer; (**b**) cyclic loading facility.

**Figure 6 materials-12-00716-f006:**
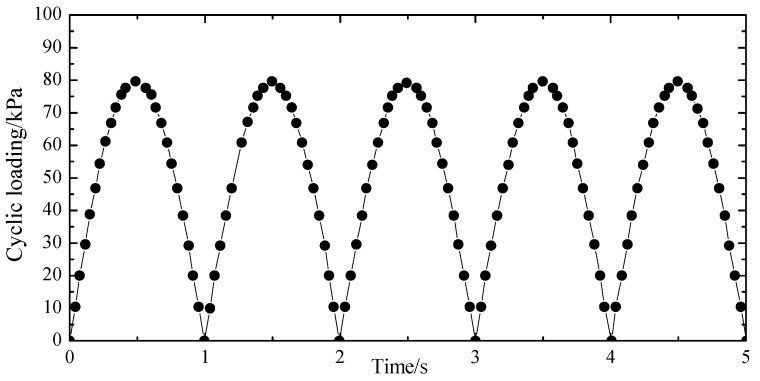
Loading waveform.

**Figure 7 materials-12-00716-f007:**
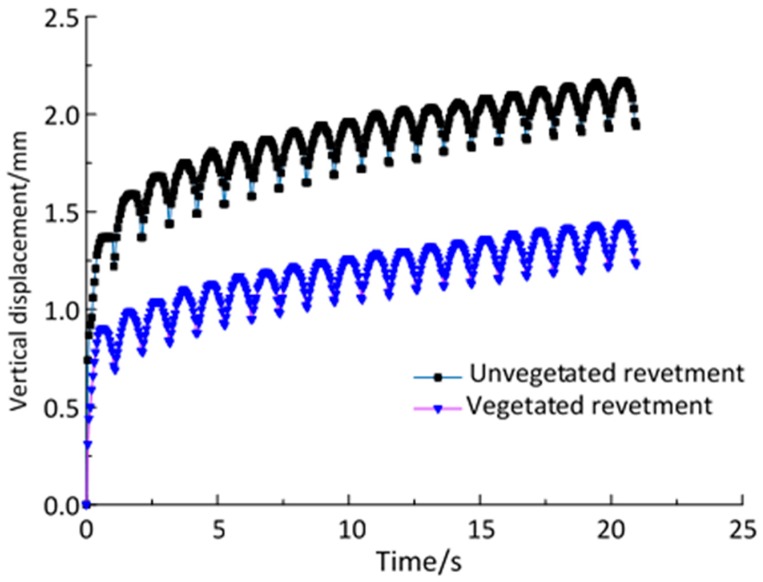
Vertical displacement in the initial 20 s.

**Figure 8 materials-12-00716-f008:**
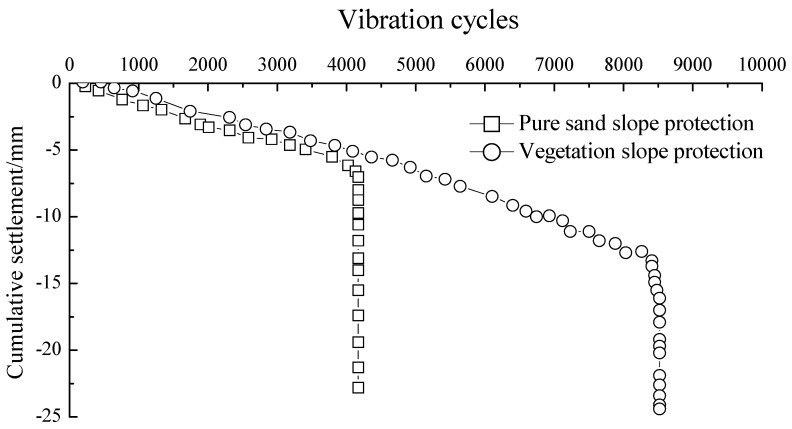
Vertical displacements during the loading process.

**Figure 9 materials-12-00716-f009:**
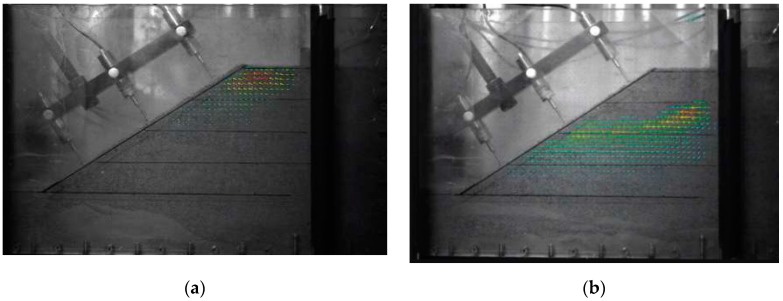
Displacement vectors at failure adopting the particle image velocimetry (PIV) method: (**a**) vegetated revetment; (**b**) unvegetated revetment.

**Figure 10 materials-12-00716-f010:**
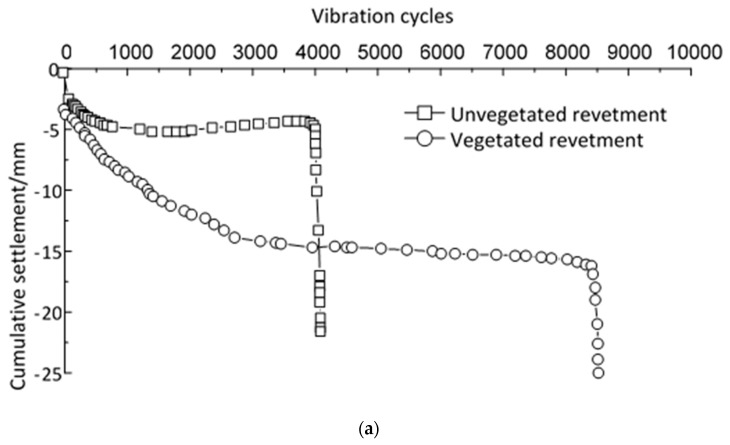
Cumulative deformation curve of slopes: (**a**) revetment slope crest; (**b**) middle of revetment slope; (**c**) top of revetment slope.

**Figure 11 materials-12-00716-f011:**
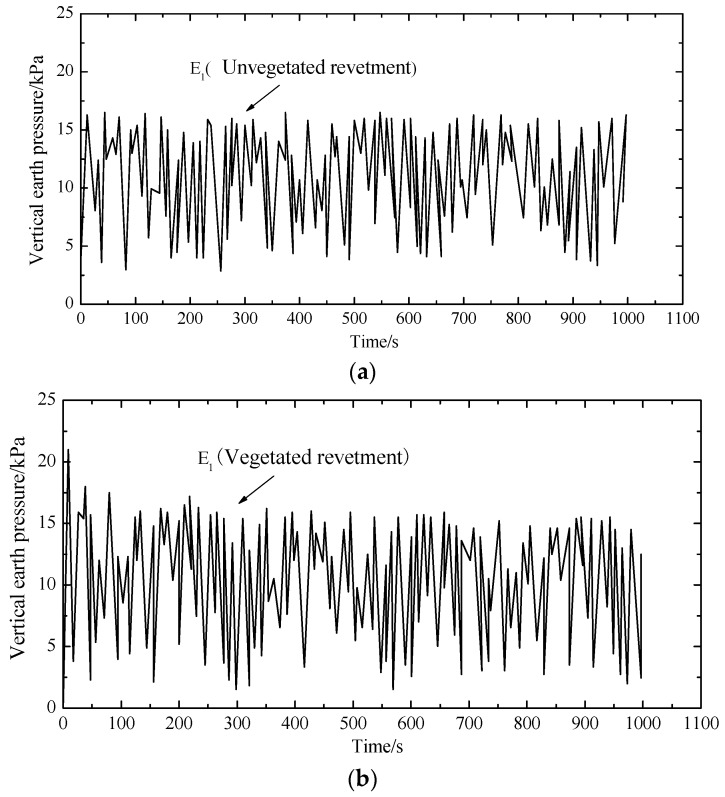
Dynamic earth pressure at E1: (**a**) unvegetated revetment slope (E1); (**b**) vegetated revetment slope (E1).

**Figure 12 materials-12-00716-f012:**
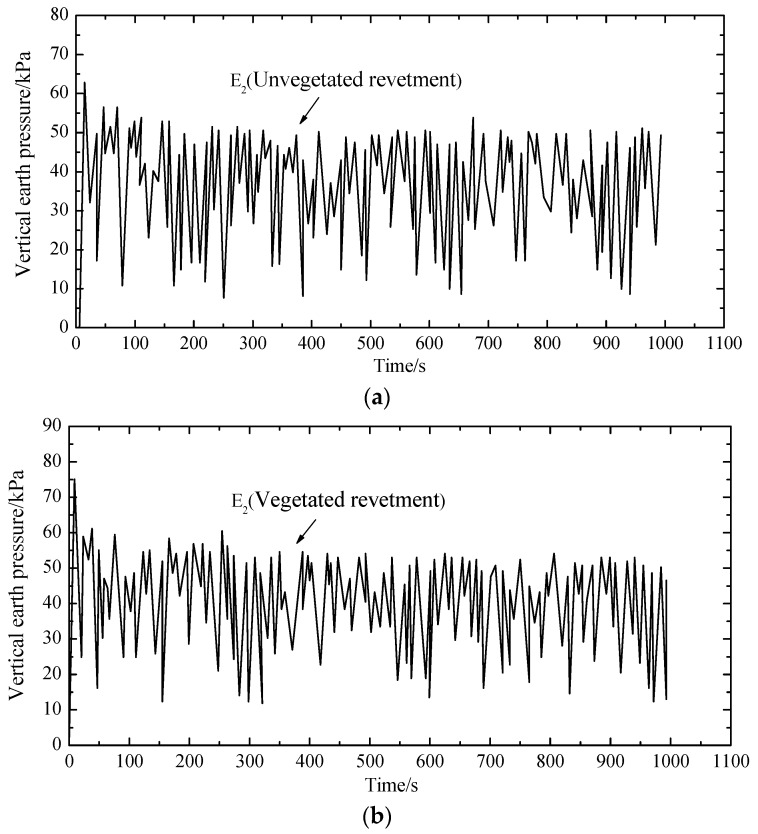
Dynamic earth pressure at E2: (**a**) unvegetated revetment slope (E2); (**b**) vegetated revetment slope (E2).

**Figure 13 materials-12-00716-f013:**
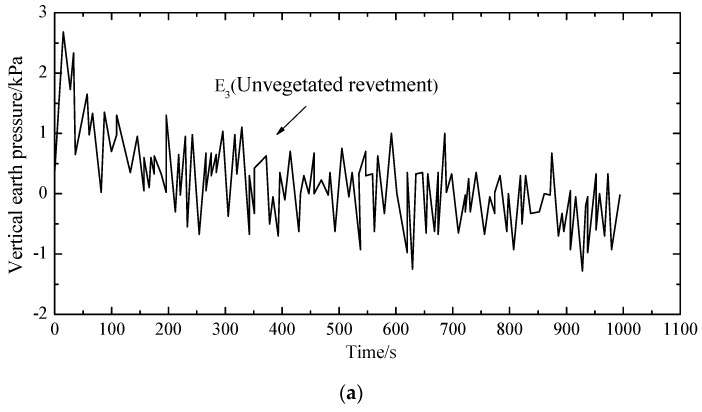
Dynamic earth pressure at E3: (**a**) unvegetated revetment slope (E3); (**b**) vegetated revetment slope (E3).

**Figure 14 materials-12-00716-f014:**
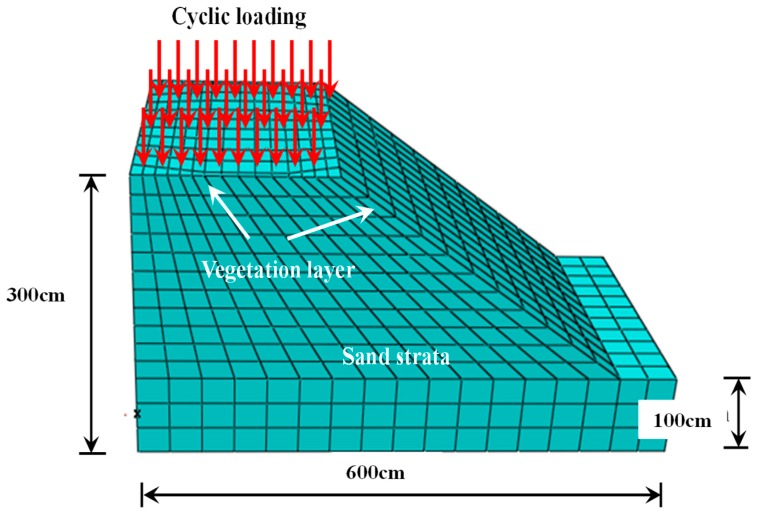
Three-dimensional finite element model.

**Figure 15 materials-12-00716-f015:**
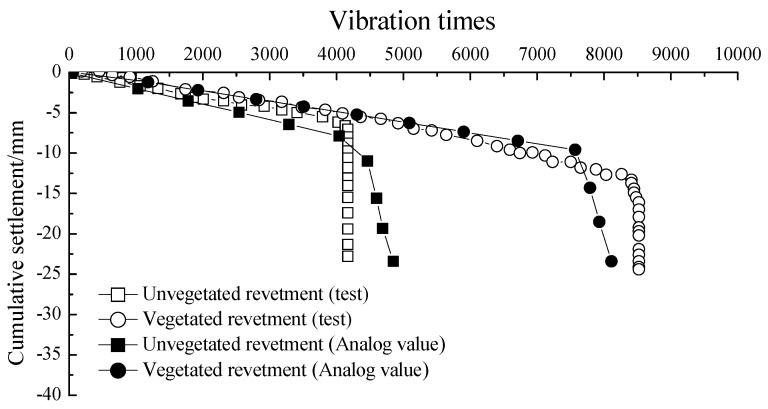
Comparison of cumulative vertical settlement results between experimental tests and numerical simulations.

**Figure 16 materials-12-00716-f016:**
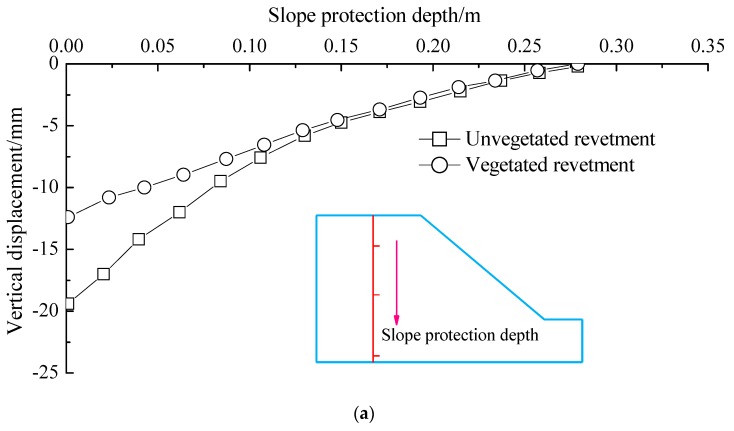
Displacement variations along the revetment depth: (**a**) vertical displacement; (**b**) lateral displacement.

**Figure 17 materials-12-00716-f017:**
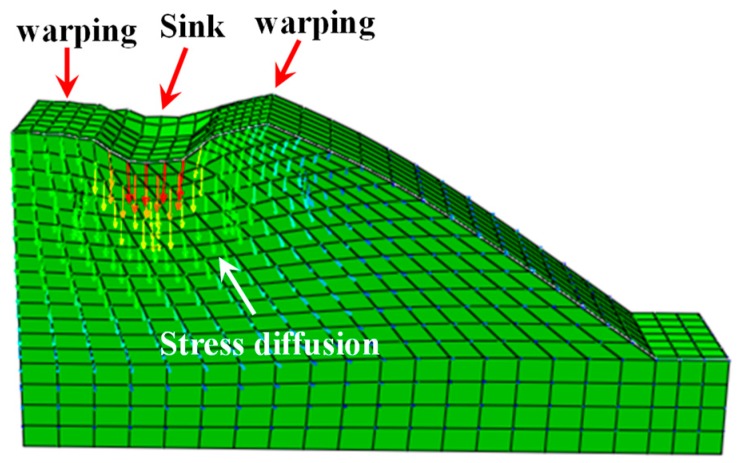
Sliding trend of vegetation slope protection.

**Figure 18 materials-12-00716-f018:**
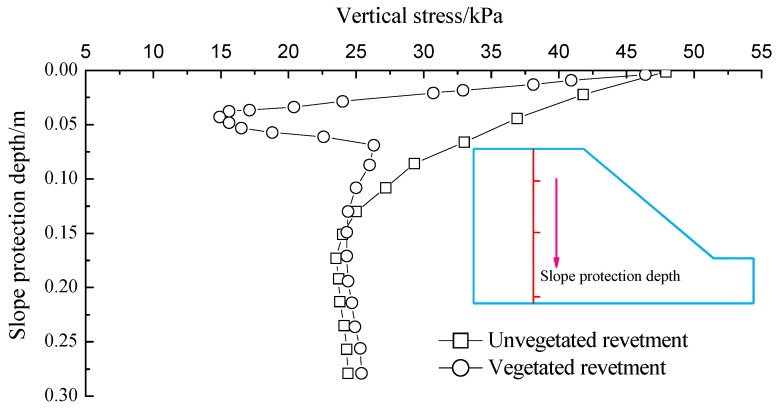
Vertical stress along the revetment depth.

**Table 1 materials-12-00716-t001:** Properties of vegetated bituminous mixture.

Porosity (%)	Compressive Strength Under 15 °C (MPa)	Compressive Modulus Under 15 °C (MPa)	Dynamic Stability Under 60 °C (Number/mm)	Total Deformation (mm)	Three-Day Flying Loss (%)	Permeability Coefficient (cm/s)
23	2.19	412.12	2875	2.90	12.7	0.039

**Table 2 materials-12-00716-t002:** Physical and mechanical properties of the foundation soil and bituminous mixture.

Raw Material	*µ*	*E*/MPa	*c*/kPa	*φ*/(°)	*ρ*/g.cm^−3^	Moisture (%)
Foundation soil	0.32	10.82	2.2	25	1.91	35.7
Asphalt mixture	0.27	395	—	—	1.92	23.52

**Table 3 materials-12-00716-t003:** Monitoring results of soil pressure.

Measurement Point	Un- Vegetated E1	Vegetated E1	Un- Vegetated E2	Vegetated E2	Un- Vegetated E3	Vegetated E3
Mean value/kPa	9.57	9.49	34.02	37.68	0.23	0.30
Maximum value/kPa	15.79	18.65	57.45	67.43	3.32	0.95

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
