# Peer review of "Dynamic Characteristics and Failure Mechanism of Vegetated Revetment under Cyclic Loading"

_materials, 2019, doi:10.3390/ma12050716_

Reviewer 1 Report

This paper studies of the failure mechanism of ecologically vegetated bituminous mixture. For this purpose, experimental studies on dynamic characteristics of asphalt mixtures along with elastic-plastic finite element modeling were conducted. Comparisons between the results of vertical settlement were made.
This is an interesting paper. However, this reviewer does not recommend the publication of the manuscript in the present form because of the following reasons:

1) This reviewer think it will be useful if the authors provide some additional information on the validation of tests used in this study.

2) English usage and spelling should be improved.

3) The manuscript needs more detailed description of the methodology and analysis of test results.

4) The manuscript is focused on the elastic-plastic properties of asphalt mixture and needs more detailed information on the state of the art study. Such as viscoelastic properties presented in the following references:

·         Bazzaz, M., Darabi, M. K., Little, D. N., & Garg, N. (2018). ”A straightforward procedure to characterize nonlinear viscoelastic response of asphalt concrete at high temperatures.” Transportation Research Record, 2672(28), 481-492.

This paper will recommend for publication if the authors consider the above suggestions to improve the quality of the manuscript. Some editing is still needed. I believe it will be done before publishing.

Author Response

Reviewer#1

This paper studies of the failure mechanism of ecologically vegetated bituminous mixture. For this purpose, experimental studies on dynamic characteristics of asphalt mixtures along with elastic-plastic finite element modeling were conducted. Comparisons between the results of vertical settlement were made.

This is an interesting paper. However, this reviewer does not recommend the publication of the manuscript in the present form because of the following reasons:

Response:

The authors thank the reviewer for all of these positive comments. All comments are addressed now below and correspondingly in the revised manuscript.

This reviewer think it will be useful if the authors provide some additional information on the validation of tests used in this study.

Response:

The authors have tried the best to add more details for the validation tests. For example, under the revised Section 4.2.,

“These geotechnical parameters were obtained from the prior physical tests, including the exact values of cohesion and internal friction angle shown in Table 2, with Poisson ratio at 0.3.” ;

“The bottom of the slope was fully constrained, and horizontal constraints were applied to the left and right sides of the slope. The ideal elastoplastic constitutive model associated to the flow law was applied to the soil.”;

Following Fig.16, it is now added that

“Therefore, the vegetative technique has limited reinforcing effects on slopes prone to deep-level damage. Such type of slopes should be first strengthened in other approaches before the vegetation protection approach is adopted.”

Following Fig. 18, it is now added that

“The strength of the shallow soil increases with the root content in the soil.”

2) English usage and spelling should be improved.

Response:

The English usage and spelling have now again been double-checked and corrected of any errors, as can be seen in the change-tracked version of the revised manuscript.

3) The manuscript needs more detailed description of the methodology and analysis of test results.

Response:

More detailed descriptions in the methodology section are provided under Section 2. For example, under Section 2.1.1, “The original bituminous mixture was firstly compressed into desired forms using machines in the factory. The compressed and shaped mixture samples were then cooled down before the porous asphalt was infiltrated with the cultivation solution.”;

“The cultivation solution should be abundant to immerse the asphalt samples. Generally the porous asphalt has a porosity rate around 20% and could be fully filled with cultivation solution theoretically. But practically, generally by volume between 10% and 15% of asphalt samples were filled by cultivation solution.”;

An extra table (Table 1) is now added to show the properties of vegetated bituminous mixtures.

Table 1. Properties of vegetated bituminous mixture

Porosity

(%)

Compressive   strength under 15℃ (MPa)

Compressive   modulus under 15℃ (MPa)

Dynamic   stability under 60℃ (number /mm)

Total   deformation (mm)

3 Day   Flying loss(%)

Permeability   coefficient (cm/s)

23

2.19

412.12

2875

2.90

12.7

0.039

Under Section 2.3., it is now also added that:

“When the slope is low, the soil on the revetment can be stabilized even in the unreinforced state; if the revetment is too steep, the stability of the soil will be poor, and a slide could easily occur. In this research, the test facility shown in Fig.4 with a slope of 35° was decided and built.”

4) The manuscript is focused on the elastic-plastic properties of asphalt mixture and needs more detailed information on the state of the art study. Such as viscoelastic properties presented in the following references:

Response:

Thanks for the reviewer’s suggestion. Now more suggested references have been added in the revised manuscript, including:

[22] Bazzaz, M., Darabi, M. K., Little, D. N., & Garg, N. (2018). ”A straightforward procedure to characterize nonlinear viscoelastic response of asphalt concrete at high temperatures.” Transportation Research Record, 2672(28), 481-492.

[20] Zhou, J.; Chen, X.; Fu, Q.; Xu, G.; Cai, D. Dynamic Responses of Asphalt Concrete Waterproofing Layer in Ballastless Track. Appl. Sci. 20199, 375.

[21] Si, C.; Cao, H.; Chen, E.; You, Z.; Tian, R.; Zhang, R.; Gao, J. Dynamic Response Analysis of Rutting Resistance Performance of High Modulus Asphalt Concrete Pavement. Appl. Sci. 20188, 2701.

This paper will recommend for publication if the authors consider the above suggestions to improve the quality of the manuscript. Some editing is still needed. I believe it will be done before publishing.

Response:

The authors thank the reviewer for the generally positive comments. All the above-mentioned suggestions are now adopted, and the editing work has been performed.

Reviewer 2 Report

Dear authors,

thank you for such unusual why of pours asphalt application.

However, there are some uncertainties:

What percentage of cultivation solution and porous asphalt contains in vegetated bituminous mixture?

Porous asphalt is layered at 150°C and compacted with roller compactor, how this vegetated bituminous mixture is going to be constructed (layered) and compacted? Does the seeds of plans will not be affected by the asphalt mixture heat? Also, it will be very difficult to construct and compact the vegetated bituminous mixture in 35° slope of riverbank revetment.

The experiment evaluate idealistic conditions, the hydrogeological (moisture) condition variation in riverbank soil during the year. It was not clear what actual size the experiment. In line 125 it is written that “the thickness of 750px for each soil layer” was used to ensure density and moisture conditions. But in figure 4 (line 172) it is shown, that total thickness of modelled structure is 300 mm, which is 30 cm.

In the table 1 (159) it is presented properties for soil and bituminous, but properties for vegetated bituminous mixture was not shown. Under the table (line 161) is written that for unvegetated and vegetated slopes the compaction ratio and the void ratio is different. But the actual mechanical properties of vegetated bituminous mixture had to be tested in the laboratory, now it seems to be somehow assumed.

Also, the plant root influence for vegetated bituminous mixture stiffness was investigated only under short term it is not known, how stiffness will change with the year, when the root density will increase, and bituminous part of mixture will aged. 

The idea of vegetated bituminous mixture is interesting, just more experimental test have to be done before actual application.      

Author Response

What percentage of cultivation solution and porous asphalt contains in vegetated bituminous mixture?

Response:

The mixing of cultivation solution is explained under Section 2.1.1, e.g., the porous asphalt was immersed in the cultivation solution. The cultivation solution should be abundant to immerse the asphalt. Generally the porous asphalt has a void rate around 20% and could be fully filled with cultivation solution theoretically. But practically, generally by volume between 10% and 15% of asphalt samples were filled by cultivation solution. 

Porous asphalt is layered at 150°C and compacted with roller compactor, how this vegetated bituminous mixture is going to be constructed (layered) and compacted? Does the seeds of plans will not be affected by the asphalt mixture heat? Also, it will be very difficult to construct and compact the vegetated bituminous mixture in 35° slope of riverbank revetment.

Response:

The authors appreciate the reviewer’s concerns. More details are now provided in the revised manuscript as described below:

The original bituminous mixture was firstly compressed into desired forms using machines in the factory. The compressed and shaped mixture samples were then cooled down before the porous asphalt was infiltrated with the cultivation solution.

P.S.: the authors have obtained the China National Patent (No. ZL201410212432.9) of cooling down the compressed and well-formed bituminous mixture sample.

Regarding the slope of riverbank, more details are now added in the revised Section 2.1.2 following Fig.2, e.g., the vegetated bituminous mixture was placed evenly on the revetment surface after the revetment was previously formed and properly saturated.

The experiment evaluate idealistic conditions, the hydrogeological (moisture) condition variation in riverbank soil during the year. It was not clear what actual size the experiment. In line 125 it is written that “the thickness of 750px for each soil layer” was used to ensure density and moisture conditions. But in figure 4 (line 172) it is shown, that total thickness of modelled structure is 300 mm, which is 30 cm.

Response:

The author confirms that there is no 750x mentioned regarding the soil layer, but it is 300mm or 750px, and that is consistent with Fig.4.

In terms of ideal conditions, the authors admit the limitations in the end of the manuscript, e.g., “The physical and numerical modeling in this study did not consider other influence factors to river bank conditions, such as the waving effects on erosion, the changing soil-water pressures, or saturation conditions during the high levels of open water. Future tests of dynamic characterics of vegetated revetment could extend the current research findings by introducing these influence factors.”

In the table 1 (159) it is presented properties for soil and bituminous, but properties for vegetated bituminous mixture was not shown. Under the table (line 161) is written that for unvegetated and vegetated slopes the compaction ratio and the void ratio is different. But the actual mechanical properties of vegetated bituminous mixture had to be tested in the laboratory, now it seems to be somehow assumed.

Response:

The authors mention the properties of vegetated bituminous mixture under Section 2.1.2 following Fig 2., by referring to the authors’ own earlier studies [32].

[35] Chen, W., Gen, J., Xu Y.D. A Study on Planting Property Test of Environmental Ecotype Asphalt Mixture. Shanghai Environmental Science. 2015,34(3), 127-131.

But to further address the reviewer’s concern, an extra table is now added in the revised manuscript showing the properties of the vegetated bituminous mixture.

Table 1. Properties of vegetated bituminous mixture

Porosity

(%)

15℃ Compressive strength (MPa)

15℃Compressive modulus (MPa)

60℃ Dynamic   stability (time/mm)

Total deformation (mm)

3 Day Flying loss(%)

Permeability coefficient (cm/s)

23

2.19

412.12

2875

2.90

12.7

0.039

Also, the plant root influence for vegetated bituminous mixture stiffness was investigated only under short term it is not known, how stiffness will change with the year, when the root density will increase, and bituminous part of mixture will aged. 

 Response:

The authors appreciate the reviewer’s concern. The suggestion of longer-term durability during the vegetated bituminous mixture is not the focus of the current study. But the reviewer provides a very good direction to continue the current work in the future by considering the plant root influence in the long term perspective. It is now added in the end of the conclusion to propose this future research direction, in that “Another recommended research for future work would be to investigate the plant root influence on the vegetated bituminous mixture by considering the long-term stiffness, root density, and the aging bituminous part of the mixture.”

The idea of vegetated bituminous mixture is interesting, just more experimental test have to be done before actual application.

Response:

The authors thank the reviewer for the generally positive comments. These experimental findings serve as the initial scientific work before the future application in the field. More limitations and recommendations have been explained in the end of the conclusion, such as “Future tests of dynamic characteristics of vegetated revetment could extend the current research findings by introducing these influence factors. For example, it is known that under the condition of soil saturation, with the increase of water content, the soil cohesion and internal friction angle of the phyto sanitary layer would decrease. But the embedded effect of vegetation roots still exists to protect the river bank slope by constricting the horizontal displacement. However, whether and how the embedding effect will change due to soil saturation require further research.”

Round  2

Reviewer 1 Report

There are room to improve.